# YOLOv8n-GSS-Based Surface Defect Detection Method of Bearing Ring

**DOI:** 10.3390/s25216504

**Published:** 2025-10-22

**Authors:** Shijun Liang, Haitao Xu, Jingyu Liu, Junfeng Li, Haipeng Pan

**Affiliations:** 1School of Information Science and Engineering, Zhejiang Sci-Tech University, Hangzhou 310018, Chinaljf2003@zstu.edu.cn (J.L.); 2Changshan Research Institute, Zhejiang Sci-Tech University, Quzhou 324299, China

**Keywords:** bearing rings, target detection, YOLO, surface defect detection

## Abstract

Industrial bearing surface defect detection faces challenges such as complex image backgrounds, multi-scale defects, and insufficient feature extraction. To address these issues, this paper proposes an improved YOLOv8-GSS defect detection method. Initially, the network substitutes the standard convolution in the C2f module and Concat module within the neck module with lightweight convolutions, GsConv, thereby reducing computational costs. Subsequently, to better capture and represent crucial features in the images, an SENetV2 attention mechanism is integrated before the SPPF module at the backbone end, effectively enhancing the model’s accuracy and robustness in defect detection. Finally, a self-built dataset of surface images of bearing rings collected from industrial sites is utilized as the basis for extensive experimentation. Experimental results show that the network achieves 97.8% AP50, with detection accuracy for large-, medium-, and small-scale defects improved by 2.4%, 3.6%, and 2.3%, respectively.2.3% respectively. The detection speed reaches 115 frames per second (FPS). Compared to mainstream surface defect detection algorithms, the proposed method exhibits significant improvements in both accuracy and detection speed.

## 1. Introduction

As a crucial component within modern industry, bearings are regarded as the “joints” of mechanical devices. The primary function of bearings is to support rotating shafts or other moving bodies, and to reduce the coefficient of friction and secure the rotating shaft during mechanical transmission processes [1]. With the continuous advancement of computer software and hardware technologies, the introduction of machine vision inspection systems into production sites has become feasible. Through the utilization of digital image techniques, effective defect detection and classification of bearing rings can be achieved, thereby enhancing production efficiency, reducing inspection time, and ensuring the quality compliance of bearings. This contributes to enterprise transformation and upgrading, thereby enhancing economic benefits.

In recent years, the application of machine vision and deep learning technologies in industrial defect detection has continued to expand, achieving remarkable results in multiple important fields such as glass [2,3], ceramics [4,5,6], textiles [7,8], healthcare [9,10,11], and steel [12,13]. These methods, leveraging their efficiency and automation capabilities, are gradually replacing traditional manual inspection, thereby enhancing detection accuracy and economic benefits. However, the application of vision-based intelligent detection techniques remains relatively limited in the domain of bearing surface defect inspection.

Currently, in the bearing manufacturing industry, small- and medium-sized enterprises still primarily rely on manual defect inspection. In contrast, large-scale bearing manufacturers have gradually adopted two mainstream approaches for surface defect detection: one based on traditional machine vision techniques, and the other employing deep learning-based recognition methods.

In the context of traditional machine vision-based detection, Zhou et al. [14] proposed an automatic machine vision-based method for detecting defects on bearing ring surfaces. The process includes image acquisition, preprocessing, ROI extraction, and defect recognition, implemented through a multi-station visual inspection system. However, the method struggles to detect extremely small or excessively large defects. Wang et al. [15] proposed a surface defect detection method for bearing outer rings based on machine vision. The detection process involves image acquisition using a CMOS industrial camera, gray histogram analysis, morphological background processing, Otsu segmentation, and morphological edge detection. Yet this method may be affected by the interference of complex backgrounds. Gu et al. [16] proposed an automated defect detection approach incorporating gamma correction preprocessing and a modified Canny algorithm with adaptive thresholding for segmentation. Despite its innovative integration of traditional image processing techniques, the method achieves a relatively limited recognition accuracy of 93.3% and shows reduced efficacy on surfaces with complex textures. Liu et al. [17] proposed an automatic detection system for micro-defects on bearing surfaces using self-developed hardware and software. Utilizing dual-light illumination, the system performs threshold segmentation, contour extraction, and positioning recognition to efficiently identify four types of defects (gaps, stains, shrunken lids, scratches). Nevertheless, the method requires strictly controlled lighting conditions.

In terms of deep learning-based detection schemes, Yao et al. [18] developed a rider optimization-driven convolutional neural network (RO-MCNN) for vision-based bearing surface defect detection. The method utilizes Gaussian denoising, LBP feature extraction, and MCNN classification, but demands substantial computational resources for optimization. Liang et al. [19] developed a lightweight network (LSK-YOLOv8n) with multi-level attention fusion for elastomeric bearing damage detection. The method integrates Large Selective Kernel attention and Focal-EIoU loss to handle structural interference and sample imbalance, though computational efficiency requires further optimization. Tayyab et al. [20] presented a hybrid–ensemble method for bearing defect diagnosis, combining deep features from CNNs and handcrafted features (HOG/LBP) with classical ML models. The method utilizes decision-level fusion to reduce computational cost while maintaining high accuracy. Han et al. [21] proposed BED-YOLO for bearing surface defect detection. They designed the IFC module to generate and normalize the spatial attention map, giving priority to retaining key features. In addition, by incorporating the EFFSC module to efficiently fuse features, the expressiveness and processing speed of the model were significantly improved. Li et al. [22] proposed a bearing surface defect detection method based on the improved YOLOv8. They adopted FasterNet to optimize the backbone network and combined it with BiFPN to improve the efficiency of feature fusion at different scales. Meanwhile, the SAConv module was introduced to enhance the model’s feature extraction and processing capabilities, thus improving the detection accuracy of the model. Liu et al. [23] proposed an improved RT-DETR model for bearing defect detection. They introduced the Dysample dynamic upsampling method to reduce the loss of feature information. Additionally, they incorporated the EMO module and the D-LKA attention module to enhance the model’s performance in feature extraction and feature fusion, achieving a balance between detection accuracy and detection efficiency. Zhou et al. [24] proposed a shifted-window Transformer (sSwin) for bearing defect detection. They designed shunted large–small windows to replace the shifted-window mechanism. Meanwhile, the proposed local connection module enhanced the boundary interaction between adjacent tokens, bringing significant performance gains for bearing defect detection.

The application prospects of target detection technology based on deep learning theory in surface defect detection of bearing rings are promising. YOLOv8, as one of the most advanced single-stage object detection networks currently available, stands out in the field of object detection due to its exceptional detection accuracy and efficient inference speed. This model demonstrates significant advantages in handling small object detection and dense object detection tasks, while also exhibiting strong robustness, enabling it to effectively adapt to complex and variable detection environments. Compared to subsequent versions such as YOLOv11, YOLOv8 has undergone more extensive validation and in-depth optimization in practical application scenarios, with its performance stability and practicality being thoroughly verified. Additionally, YOLOv8 benefits from a large and active technical community, supported by comprehensive and detailed documentation and resources. This provides substantial convenience for developers during model development, debugging, and optimization, allowing them to quickly access the necessary technical support, thereby significantly enhancing development efficiency and project progression. However, despite YOLOv8′s excellent performance in numerous fields, its direct application to bearing surface defect detection tasks still faces certain challenges, such as insufficient sensitivity to minute defects and a relatively high false detection rate in complex backgrounds. Further research and optimization are required to adapt the model to the specific requirements of such scenarios. The background texture of bearing rings is complex, the defect sizes are varied, and there are numerous types of defects with uneven brightness. Additionally, the production environment of bearing rings often contains a large amount of oil and dust, which can interfere with bearing ring images. Considering these challenges and the optical characteristics of bearing rings, imaging features of defects, and detection requirements, an improved YOLOv8n-GSS defect detection network is proposed, and an automatic detection system for surface defects on bearing rings is developed, achieving and realizing industrial applications. The main contributions of this paper are as follows:Replaced standard convolutions between the C2f module and Concat module in the neck module with lightweight convolutions, GsConv, to reduce computational costs.Added an SENetV2 attention mechanism before the SPPF module at the backbone end, which better captures and represents important features in the image, effectively improving the accuracy and robustness of the model for defect detection.Developed a smart visual online inspection system for surface quality of bearing rings.

The rest of this paper is organized as follows: Section 2 introduces the smart visual online inspection system for surface quality of bearing rings. Section 3 details the detection method, including the network structure of improved YOLOv8. Section 4 presents the experimental validation of our method. Section 5 concludes our work.

## 2. Intelligent Visual Online Inspection System for Bearing Ring Surface Quality

### 2.1. Inspection System Introduction

The overall structure of the intelligent visual inspection system for surface defects of bearing rings designed and developed in this study is shown in Figure 1. The system consists of three major modules: mechanical transmission, machine vision, and electrical control. The mechanical transmission module, based on a frame structure, utilizes motor drives and pneumatic cylinders to achieve precise grasping and positioning of the bearings, transporting them steadily to designated inspection stations. The machine vision module integrates high-resolution industrial cameras, dedicated optical lighting systems, industrial computers, and specialized visual detection algorithms. It performs high-quality image acquisition of bearing surfaces at various workstations, providing a reliable data foundation for subsequent defect identification and classification. The electrical control module, centered on a Programmable Logic Controller (PLC) and high-precision photoelectric sensors, is responsible for system operational logic control and real-time signal interaction, ensuring coordination within the inspection process and responsiveness.

In the design of the visual system, the MV-CS200-10GM line scan camera from the Chinese company Hikvision (Hikvision, Hangzhou, China) was selected as the primary imaging device. This camera features a resolution of 20 megapixels, capable of covering the entire bearing area, thus providing a high-resolution foundation for subsequent imaging. Furthermore, during the imaging process, a high-resolution MVL-KF1628M-12MP lens, also from Hikvision, and a dome-shaped shadowless light source were equipped to achieve clear imaging of the bearing ring surface. Such a combination ensures the quality and accuracy of imaging, providing reliable data support for subsequent defect detection. Finally, a detection prototype was constructed, with its exterior appearance depicted in Figure 1a, while the internal structure can be observed in Figure 1b, and the lighting arrangement utilizing the camera and light source combination is illustrated in Figure 1c.

The industrial assembly line imposes a throughput requirement of 6–10 s per piece for the automatic defect detection system of bearing inner rings. To objectively evaluate the deployment suitability of different models, this study conducts experiments in a hardware environment identical to that used in industrial settings. Through key performance metrics such as frames per second (FPS), it systematically investigates whether each model can meet the real-time detection requirements under the specified production cycle.

### 2.2. Types of Bearing Ring Defects

The defects of bearing rings include Helix marks, forging waste, black spots, dents, and scratches [25]. Different types of bearing defects are illustrated in Figure 2.

1.Helix marks.

The Helix mark defect, as depicted in Figure 2a, arises from sharp burrs on grinding wheels, guide wheels, and cutting plates, resulting in spiral-shaped incisions resembling blade-like points on the outer diameter of the roller. This defect typically appears black and exceeds a size of 0.5 × 1.0 mm^2^.

2.Forging waste.

As shown in Figure 2b, the forging waste defect manifests as residual unremoved black areas on the surface due to non-uniform grinding during the polishing process. This defect typically appears as dark-colored arc-shaped features.

3.Black spots.

As shown in Figure 2c, the black spot defect is characterized by the presence of irregular black spots of varying sizes on the bearing end face, with individual dimensions exceeding 0.3 × 0.3 mm^2^.

4.Dent.

As illustrated in Figure 2d, the dent defect primarily results from inevitable collisions with hard objects during handling and transportation, leading to localized surface depressions.

5.Scratches.

The scratch defect, as illustrated in Figure 2e, primarily consists of surface linear scratches of a certain depth on workpieces generated during machining, handling, and measurement processes. This defect typically appears black and exceeds a size of 0.1 × 0.1 mm^2^.

6.Abrasion.

The abrasion defect is illustrated in Figure 2f. It is caused by contact and relative motion between the bearing surface and harder foreign particles, which gouges the material surface, resulting in linear scratches of varying morphology and dimensions.

### 2.3. Detection Difficulties

Based on the imaging characteristics and detection requirements of bearing ring defects, the main challenges are as follows:1.Variety of defect types.

The defects of bearing rings are distributed on end faces, side faces, raceways, and flanges. The production process is complex, resulting in diverse defect morphologies and complex texture backgrounds.

2.Susceptible to interference.

The production workshop environment for bearing rings is complex and variable, subject to interference from factors such as dust, oil stains, and temperature changes. Dust on the bearing ring surface can occur, and the imaging characteristics of dust are very similar to those of black spot defects, leading to potential misjudgments.

3.Similar defects.

The forging waste defect is very similar to the black spot defect, making it difficult to differentiate between them.

4.Varied sizes.

Defects such as spiral threads, forging waste, and scratches vary significantly in size. Therefore, the detection model needs to effectively detect targets of different scales.

5.Different lighting requirements.

Different defects have different requirements for the brightness of the light source. For example, dent defects may not be identifiable under high brightness, while forging waste defects may be missed under low brightness.

## 3. Bearing Ring Defect Detection Algorithm Based on YOLOv8-GSS

### 3.1. YOLOv8 Introduction

YOLOv8, introduced by Glenn Jocher et al., represents a significant advancement in the YOLO (You Only Look Once) family of object detection architectures, building upon the design principles of earlier versions such as YOLOv3 and YOLOv5. Its structure includes input preprocessing modules, a convolutional backbone for feature extraction, a feature fusion network based on a path aggregation network (PAN), and three decoupled detection heads for improved classification and regression. The model incorporates several enhancements such as anchor-free detection and a redesigned loss function, contributing to its strong performance at the time of release.

YOLOv8 employs the same data preprocessing strategy as YOLOv5, incorporating four augmentation techniques during training, including Mosaic augmentation, Mixup augmentation, random perspective transformation, and HSV color augmentation. YOLOv8 adopts the CSPDarknet concept from YOLOv5 as its backbone network structure and replaces the previous C3 module with a new C2f module. This new module introduces more branches, providing richer substreams for gradient backpropagation.

YOLOv8 continues to use the PAN-FPN structure to construct a feature pyramid for comprehensive fusion of multi-scale information. Additionally, it removes the 1 × 1 convolution in the CBS of the upsampling stage in YOLOv5 and replaces the C3 module with the C2f module. In contrast to the detection heads of YOLOv3 and YOLOv5, YOLOv8 has always utilized a “coupled” approach, employing two parallel branches to extract category features and position features separately. YOLOv8 employs the Task-Aligned Assigner to address the sample matching problem. This method dynamically adjusts the ratio of positive and negative sample allocations, adapting better to different tasks and data distributions.

### 3.2. YOLOv8-GSS Defect Detection Network for Bearing Rings

According to the network’s scaling factors, YOLOv8 offers five models of different scales: N/S/M/L/X. As bearing ring defect detection is an industrial application requiring consideration of both detection accuracy and speed, this paper adopts YOLOv8n as the base network architecture. By integrating GsConv convolution and the SENetV2 attention mechanism, an improved YOLOv8n-GSS defect detection network is proposed. The YOLOv8-GSS network structure, as depicted in Figure 3, enhances the detection accuracy of surface defects on bearing rings without compromising detection speed, aligning with the requirements of online detection for bearing ring surface defects.

#### 3.2.1. GSConv

In computational neuroscience, it is recognized that models with a higher number of neurons possess stronger nonlinear expressive capabilities. However, the powerful information processing abilities and low energy consumption of the biological brain cannot be overlooked, far surpassing the capabilities of computers. Therefore, constructing powerful models cannot simply rely on continuously increasing the number of model parameters.

On the contrary, adopting lightweight designs can effectively alleviate the current challenges of high computational costs. To achieve this goal, the introduction of depthwise-separable convolution (DSC) operations has yielded significant results by reducing the number of parameters and FLOPs. Figure 4a,b illustrate the computational processes of DSC and standard convolution (SC). However, it is noted that the feature extraction and fusion capabilities of DSC are far inferior to SC, which represents a limitation.

In some outstanding lightweight models such as Xception, MobileNets, ShuffleNets, etc., the speed of detectors has been significantly improved by employing depthwise-separable convolution (DSC) operations, but at the expense of lower accuracy. Although several solutions have been proposed to address this issue, such as MobileNets utilizing a large number of 1 × 1 dense convolutions to fuse independently calculated channel information, ShuffleNets employing channel shuffle to facilitate channel interaction, and GhostNet adopting halved SC operations to retain inter-channel interaction information, these methods tend to consume more computational resources. Moreover, the effectiveness of channel shuffle still falls short of the results achieved by SC.Our implementation of these models is based on the PyTorch framework (v1.10) and the timm library (v0.6.12), which provide reference implementations for these architectures.

To bring the output of DSC as close as possible to that of SC, a novel method called GSConv is introduced. As illustrated in Figure 5, “DWConv” represents the DSC operation. By utilizing shuffle to permeate the information generated by SC (dense convolution operation) into each part of the information generated by DSC, this approach ensures that the information from SC is fully blended into the output of DSC.

The advantage of dense convolution lies in its ability to preserve connections between channels to the maximum extent, thereby ensuring richer semantic information transmission, while sparse convolution may lead to the loss of partial semantic information. The design goal of GsConv is to preserve these connections to the maximum extent possible. However, employing GsConv at various stages of the model would increase the depth of the network, potentially causing data flow congestion and prolonging inference time. Therefore, a more suitable approach is to use GsConv only in the neck stage of the model.

Utilizing GsConv [26] to process concatenated feature maps in the neck stage is highly appropriate, as at this point, the channel dimension is maximal while the width and height dimensions are minimal, indicating fewer redundant pieces of information and thus requiring less compression. Consequently, better results can be achieved.

In the YOLOv8 architecture, the neck module is positioned between the backbone network and the prediction output head, playing a critical connecting role. Given the unique structure of the neck module, which encompasses both bottom-up and top-down integration capabilities of features across various scales, it lays the foundation for subsequent predictions effectively, thereby significantly influencing the algorithm’s performance. This paper replaces the standard convolution between the C2f module and the Concat module in the neck module of YOLOv8n with the lightweight convolution method GsConv, with a computational cost approximately 60% to 70% of the standard convolution.

#### 3.2.2. SENetV2 Attention Module

An attention mechanism is a technique that assists neural network models in learning the importance of input information. Its core idea lies in assigning different weights to different parts of the input data to enhance model performance and accuracy, while also helping to prevent overfitting and strengthen model robustness. Different types of attention mechanisms include the channel attention mechanism, spatial attention mechanism, and hybrid-domain attention mechanism. The channel attention mechanism aims to assess the importance of each channel in the feature map and allocate weights accordingly to guide the model to focus more on crucial channels. A representative method of this approach is SENet [27]. On the other hand, the spatial attention mechanism enhances the model’s understanding of spatial information by generating and scoring masks for space. The representative model for this approach is SAM. The hybrid-domain attention mechanism integrates the advantages of both channel and spatial attention, such as BAM and CBAM.

SENetV2 [28] is an improved version of SENet, aiming to further enhance feature representation effectiveness. Compared to SENet, SENetV2 has been optimized in several aspects. Firstly, SENetV2 introduces more modules and mechanisms to enhance the network’s capture of channel patterns and understanding of global knowledge. Secondly, SENetV2 adopts a more complex structure and more parameters to increase the network’s representation capacity and flexibility. Additionally, SENetV2 further improves the model’s training efficiency and convergence speed through carefully designed network architectures and training strategies. Overall, SENetV2 not only improves feature representation effectiveness but also maintains good computational efficiency and practicality, making it one of the important research directions in the current field of object detection.

Figure 6a illustrates the ResNext aggregation module, which consists of a multi-branch CNN module. Figure 6b presents the SE module, comprising operations for compressing the input and then restoring its original shape. In the compression operation, the downsized input is not fed back to restore the original shape. Figure 6c showcases the SENetV2 module, which combines components from (a) and (b), consisting of multi-branch fully connected layers.

Figure 7 illustrates the internal functionality of the proposed SaE module of SENetV2. This module takes the output of the squeeze operation and feeds it into multi-branch fully connected layers, followed by an activation process. The split input is then passed to the final step to restore its original shape.

Figure 8 presents a comprehensive comparison between the SENet module and the SENetV2 module. In the figure, smaller rectangles represent fully connected (FC) layers or dense layers, while convolutional layers are denoted as “Conv”. The squeeze operation and subsequent activation or aggregate activation operations are always combined with the input to ensure restoration of the original shape. Multi-branch dense layers are connected within circles.

SENetV2 introduces a novel aggregated multi-layer perceptron, termed multi-branch dense layers, incorporating the design of compression–excitation residual modules to surpass existing architectures. This approach achieves better feature representation by enhancing the network’s capture of channel patterns and understanding of global knowledge. Compared to SENet, the proposed model hardly increases parameters. By utilizing aggregation modules containing multiple convolution operations to handle branch inputs instead of opting for deeper networks or wider layers, the model can more effectively capture complex spatial representations, particularly within the convolution layers of the aggregation modules.

The reasons for incorporating the SENetV2 module before the SPPF module in YOLOv8n in this study are as follows: (1) This module can more effectively capture and showcase key features in images by enhancing the network’s capture of channel patterns and understanding of global knowledge. (2) In terms of computational efficiency, the SENetV2 module outperforms other modules. With fewer computational requirements, it exhibits higher computational efficiency compared to some other attention modules.

#### 3.2.3. Loss Function

The loss function is used to measure the discrepancy between the predicted values and the ground truth values. The smaller the value of the loss function, the closer the predicted output to the expected output [29]. The loss functions employed in this study are divided into three categories: classification loss, localization loss, and confidence loss [30]. The classification loss evaluates the accuracy of assigning an anchor box to its designated class, reflecting the confidence level of category membership; the loss used to estimate the error between the predicted box and the ground truth box is referred to as the localization loss, while the confidence loss represents the network’s confidence in the presence of the target, typically ranging from 0 to 1, with higher values indicating a higher likelihood of the target’s presence.

The comprehensive loss function is the weighted sum of these three loss functions, as shown in Equation (1):(1)$$Loss\;={\;w}_{\mathrm{box}}L_{\mathrm{box}\;}+{\;w}_{\mathrm{obj}}L_{\mathrm{obj}}\;+\;w_{\mathrm{cls}}L_{\mathrm{cls}}$$
where the weighting coefficients are, respectively, $w_{box}$, $w_{obj}$, and $w_{cls}$.

$L_{box}$ uses CIOU as a metric, which is defined as follows, in Equation (2):(2)$$L_{\mathrm{box}}\;=\;CIoU\;=\;1\;-\;IOU\;+\;\frac{\rho^{2}\left(A,B\right)}{c^{2}}\;+\;\alpha \nu$$

IOU denotes the intersection and concurrency ratio of the predicted box to the real box, with larger values of IOU indicating that the two are closer together. ρ represents the distance between the center point of the real box A and the predicted box B, while c denotes the diagonal distance between the two enclosing boxes. The consideration of these parameters helps to characterize the relationship between the position and size of the boxes in more detail. In addition, the weighting factor α and the consistency parameter ν also play an important role in the localization loss computation. α is used to adjust the degree of influence of different parameters in the computation, while ν measures the consistency between the real frame A and the predicted frame B in terms of aspect ratio. The introduction of these parameters improves the accuracy and stability of the IOU values, leading to a better evaluation of the performance of the target detection algorithm.

IOU is defined as follows in Equation (3):(3)$$\;IOU=\frac{A\cap B}{A\cup B}$$

α and *ν* are defined as follows in Equation (4):(4)$$\alpha =\frac{\nu }{1-IOU+\nu }$$(5)$$\;\nu =\frac{4}{\pi^{2}}$$

The binary cross-entropy function is used for both classification loss and confidence loss and is defined as follows in Equation (6):(6)$$L_{\mathrm{cls}}\;={\;L}_{\mathrm{obj}\;}=\;-\frac{1}{n}\overset{\underset{n}{i=1}}{\sum }\left(y_{i}\times \ln x_{i}\;+\;\left(1\;-y_{i}\right)\times \ln\left(1\;-{\;x}_{i}\right)\right)$$
where n denotes the number of input samples, $y_{i}$ denotes the target value, and $x_{i}$ denotes the predicted output value.

## 4. Experimental Verification

### 4.1. Construction and Preprocessing of the Bearing Inner Ring Surface Defect Dataset

#### 4.1.1. Data Acquisition System

The dataset of bearing ring surface defects (DO-BSD) used in this study was constructed using a self-developed, enclosed bearing ring surface defect detection system on an actual industrial production assembly line. The system features a fully enclosed physical structure with integrated controllable lighting units, effectively eliminating interference from ambient light on defect detection and ensuring absolute consistency in imaging conditions. The dataset covers multiple types of tapered roller bearing inner rings, including models such as 30315, 30307, and 32207, spanning over ten common specifications with inner diameters ranging from 30 mm to 80 mm. All collected bearing samples comply with the ISO 355:2019 international standard [31], ensuring a unified normative basis for geometric structure and dimensions.

To guarantee the consistency and reproducibility of the image data and provide reliable input for deep learning model training, strict control measures were implemented during the data acquisition process, as detailed below:Sensing and Triggering Mechanism: The system utilizes an EE-SX671 photoelectric sensor from OMRON Corporation, Japan, to detect bearing positioning. The selection of this specific sensor model was primarily based on the inherent strong anti-interference capability and high stability of photoelectric sensors in complex industrial environments. Compared to infrared sensors, this type exhibits superior suppression of common interference sources such as oil stains and metal reflections. The sensor’s high-repeatability positioning accuracy and response time of <1 ms enable it to reliably trigger image capture within the production line cycle. Prior to deployment, the sensor underwent on-site calibration. Using a precision displacement platform, we fine-tuned its installation angle and detection distance. The calibration outcome was verified by repeatedly passing a reference bearing through the station; iterations were performed until the trigger was activated exclusively when the bearing centroid was within ±0.5 mm of the predefined location. This ensures that a trigger signal is sent to the PLC and industrial camera only when the bearing reaches the preset imaging position. This mechanism physically eliminates image background interference caused by random workpiece misalignment, thereby providing stable and consistent input conditions for subsequent deep learning models.Imaging System: Two high-resolution industrial cameras, the MV-CH050-10UM from Hikvision (China) and the MV-CS200-10GM, also from Hikvision, were employed along with a dome diffuse light source, the LTS-3DM175-B from Dongguan Leshi Automation Technology Co., Ltd. (Dongguan, China). This configuration generates uniform diffuse reflection illumination, effectively eliminating reflections from the curved metal surfaces of the bearings and interference from ambient light variations. It ensures that all images are captured under consistent lighting conditions, guaranteeing brightness uniformity and a high signal-to-noise ratio for the dataset from the source.

#### 4.1.2. Dataset Preprocessing and Composition

The original images acquired by the aforementioned data acquisition system have a resolution of 5472 × 3468 pixels, with each image occupying approximately 19 MB of storage space. Due to the complexity of the bearing ring manufacturing process, the sample distribution across different defect types exhibits significant class imbalance. To ensure the rationality of the training data, improve model training efficiency, and enhance data consistency, this study implemented a comprehensive data preprocessing pipeline, which consists of the following steps:Image Resizing and Cropping: The original images were uniformly cropped into 640 × 640-pixel patches to better suit the input requirements of the YOLO network.Manual Screening: Experienced operators visually inspected and selected patches containing defect regions for subsequent model training.Noise Reduction: A Gaussian filter with a kernel size of 3 × 3 and a standard deviation (σ) of 1.5 was applied to suppress high-frequency noise and imaging artifacts.Pixel Normalization: Pixel intensity values were linearly normalized from the original range of [0, 255] to [0, 1] to stabilize and accelerate the training process.Mosaic Data Augmentation: To address the inherent class imbalance in the dataset, the Mosaic augmentation technique was employed. This method synthesizes new training images by combining four randomly selected image patches, thereby helping to balance the sample distribution across different defect categories.Additional Augmentations: Further geometric and photometric transformations—including random rotation, random cropping, random scaling, and random brightness adjustment—were utilized. These augmentations simulate variations in defect locations and lighting intensities, thereby enhancing model robustness.

Upon completion of this preprocessing pipeline, the total number of defect samples was expanded from 3175 to 6805. This expansion enhances the model’s generalization capability and improves defect detection performance. The statistical distribution of various defect types after the expansion is summarized in Table 1, and a visual representation of the constructed dataset is provided in Figure 9.

The splitting of the dataset is an essential step prior to network training, as it directly impacts the rationality and validity of performance evaluation. To ensure an unbiased assessment, the preprocessed data was partitioned in a stratified manner based on defect categories, randomly divided into training, validation, and test sets in a ratio of 6:2:2. This random partitioning strategy ensures that all subsets share similar data distributions, thereby preventing evaluation bias and guaranteeing the fairness and representativeness of the training, validation, and testing processes. Detailed results of the dataset splitting can be found in Table 2.

### 4.2. Experimental Setup

Table 3 provides detailed information about the hardware environment and software versions used for the experiments. This information is essential for the training and validation of the experiments.

The training parameters of the network also affect the performance of the model, as shown in Table 4.

### 4.3. Performance Indices

To comprehensively evaluate the effectiveness of the proposed YOLOv8-GSS defect detection model and specifically analyze its sensitivity and detection capability for defects of different sizes, this study adopts the standard evaluation metric system from the COCO dataset. The selected metrics include the following: average precision (AP), mean average precision (mAP), precision at different Intersection-over-Union thresholds (AP_50_, AP_75_), and average precision for objects of varying scales (AP*_S_*, AP*_M_*, AP*_L_*). Furthermore, to assess the model’s complexity and inference efficiency, we also report the number of parameters (Params), floating-point operations (FLOPs), and frames per second (FPS). This comprehensive evaluation framework can objectively reflect the model’s performance across multiple dimensions—including detection accuracy, robustness, multi-scale adaptability, and computational efficiency—thereby providing a reliable assessment of its suitability for real-world industrial defect detection scenarios.

### 4.4. Convergence Analysis

Figure 10 and Figure 11 show the loss curves of these two models in terms of bounding box loss, confidence loss, and category loss. After comparative analysis, it is found that the YOLOv8n-GSS model converges slightly faster than the YOLOv8n model in terms of bounding box loss, confidence loss, and category loss, which indicates that the YOLOv8n-GSS model reaches the minimum value of the loss function faster during the training process.

In addition, the YOLOv8n-GSS model performs better in terms of both precision and recall, which implies that it has higher accuracy and better recall in the target detection task. These results indicate that the YOLOv8n-GSS model has better convergence performance and detection performance relative to the YOLOv8n model in the optimized case, and thus is more desirable in practical applications.

### 4.5. Hyperparametric Study

Hyperparameter configuration has a decisive impact on the generalization capability and final predictive accuracy of deep learning models. To systematically evaluate the rationality of the hyperparameter combinations adopted in this study, we employed an integrated strategy combining the control variable method and full factorial design. Comprehensive ablation experiments were conducted on key parameters including input image size, training epochs, batch size, learning rate, and optimizer. Table 5 details the experimental results, demonstrating the influence of these parameters on the performance of the YOLOv8n-GSS model on the bearing inner ring surface defect dataset.

As illustrated in Table 5, the control variable method was employed to evaluate the impact of different hyperparameter settings on network training outcomes. A comparison between Experiment 1 and Experiment 2 demonstrates that increasing the image size to 640 significantly improves model accuracy, which can be attributed to the retention of more detailed defect features at higher resolutions. In contrast, extending the training cycles in Experiment 3 resulted in a slight performance degradation compared to Experiment 1, indicating that the model had essentially converged after 100 epochs. Furthermore, as different optimizers exhibit varying sensitivities to learning rates, Experiments 1, 4–6, 10, and 11 were designed to compare the SGD and RAdam optimizers under commonly used learning rate configurations. The results clearly show that the SGD optimizer outperforms RAdam in this task. Additionally, Experiments 7–14 were systematically conducted to investigate the interaction between batch size and learning rate. Through this combined approach of control variables and full factorial design, it is possible to assess not only the individual effects of each hyperparameter but also the interplay between these two critical parameters, thereby providing comprehensive and reliable evidence for the selection of the configuration in Experiment 1.

From the experimental results, it can be observed that the hyperparameter set used in Experiment 1 yielded the highest detection accuracy, with the YOLOv8-GSS network achieving an mAP of 97.8%. This outcome validates the rationality of the parameter configuration proposed in this study.

### 4.6. Ablation Experiment

There are two model improvements, and in order to verify the validity of each improvement and the validity of the combination of the two improvements, ablation experiments were conducted on the improved model, with the experimental results detailed in Table 6.

Based on the ablation study results presented in Table 6, the contributions of each module to model performance can be systematically evaluated. The YOLOv8n baseline model achieves an mAP of 95.9% on the DO-BSD dataset. Upon integrating the GSConv lightweight convolution module, the mAP increases to 96.1%, while the number of parameters is reduced by 89,760 and FPS improves by 9 frames per second. This improvement primarily stems from the depthwise-separable convolutions and channel shuffling mechanism employed by GSConv, which significantly reduce model complexity while effectively preserving key defect features through enhanced feature reuse efficiency. Subsequently, the introduction of the SENetV2 attention mechanism further elevates the mAP to 97.2%, despite a corresponding 4% decrease in FPS. This performance gain originates from the attention mechanism’s adaptive recalibration of feature channels, enabling the model to concentrate on more discriminative features in defect regions. When both GSConv and SENetV2 are integrated, the model achieves the optimal balanced performance: mAP is significantly improved by 1.9 percentage points compared to the baseline, the parameter count is reduced by 73,376, and a net gain of 4 FPS is maintained. This demonstrates that the designed lightweight backbone network not only enhances feature extraction capability through feature reuse mechanisms but also improves feature utilization via attention guidance, ultimately achieving a synergistic optimization of detection accuracy and speed while reducing the number of parameters.

### 4.7. Comparison Experiment

In order to validate the effectiveness of the YOLOv8n-GSS defect detection model, this chapter conducts a comparative analysis with several mainstream detection models, including the single-stage detectors YOLOv5, YOLOv8, RT-DETR, and YOLOv11, as well as the two-stage detector Faster R-CNN. The comparative results, based on the standard COCO dataset evaluation metrics, are presented in Table 7.

Based on the experimental data presented in Table 7, the proposed YOLOv8n-GSS model demonstrates comprehensively superior performance compared to other models on the DO-BSD defect detection dataset. YOLOv8n-GSS achieves an AP50 of 97.8%, representing improvements of 14.2%, 0.5%, 1.0%, 2.6%, 0.6%, and 1.9% over Faster R-CNN, YOLOv5, RT-DETR, MobileNetv4, YOLOv11, and YOLOv8, respectively. Furthermore, systematic progress has been made in detection accuracy for large, medium, and small targets. This overall performance enhancement primarily stems from two key architectural improvements: Firstly, the replacement of standard convolutions and Concat operations with the lightweight GsConv module significantly enhances multi-scale feature fusion capability while maintaining model efficiency, leveraging its unique depthwise-separable convolutions and channel shuffling mechanism. Secondly, the integration of the SENetV2 attention mechanism prior to the SPPF module in the backbone network enables adaptive calibration of critical feature channels, effectively improving the model’s sensitivity to subtle defect features and its discriminative ability for multi-scale characteristics.

In terms of inference efficiency, the model achieves a detection speed of 115 FPS on the RTX A5000 hardware platform, with a theoretical throughput of 414,000 images per hour. When further considering the system overhead caused by image acquisition, data transmission, and system communication in practical industrial deployment (assuming a performance loss of approximately 50%), the effective throughput of the model can still be maintained at over 160,000 images per hour. This throughput capability fully meets the image acquisition and detection requirements of a production line configured with a six-camera station processing 10 bearings per minute. The experimental results indicate that the YOLOv8n-GSS model not only possesses high detection accuracy and fast inference speed but also retains a sufficient performance margin to handle fluctuations in industrial environments, demonstrating promising engineering applicability and deployment potential.

To provide an intuitive comparison of the detection performance across different models, this study randomly selected one sample from each of the six defect categories in the DO-BSD dataset for testing and visualized the results (as shown in Figure 12). The visualization results are consistent with the quantitative analysis, indicating significant differences in the performance of the models on these samples. Specifically, the Faster R-CNN architecture exhibited clear limitations in helical mark defects and dent defects. Comprehensive comparative evaluation demonstrates that the proposed YOLOv8n-GSS model delivers outstanding performance across all defect categories, with key metrics such as detection accuracy and model robustness significantly surpassing those of other compared models.

### 4.8. Network Generalization Experiment

To evaluate the generalization capability of the proposed YOLOv8-GSS model, additional comparative experiments were conducted on the Light Guide Plate Surface Defect Dataset (LGP-SDD) [32]. These experiments followed the same environmental configuration and methodology as those used for bearing inner ring defect detection. The LGP-SDD contains four types of defects—white spots, bright lines, dark lines, and surface defects—with all images resized to 416 × 416 pixels. After applying the same data augmentation strategy, the total number of effective samples was increased to 4112. The dataset was then divided into training, validation, and test sets in a ratio of 6:2:2 based on defect categories. As shown in Table 8, quantitative comparisons between YOLOv8-GSS and several mainstream algorithms were performed under this unified experimental setup. Furthermore, representative samples of each defect type were randomly selected for visual analysis, and the results are displayed in Figure 13, providing an intuitive comparison of the detection performance across different networks.

As shown by the experimental results in Table 8, the proposed YOLOv8-GSS defect detection network performs favorably on the LGP-SDD dataset, achieving a mean average precision (mAP) of 98.4%. Specifically, compared to Faster R-CNN, YOLOv5, RT-DETR, MobileNet, YOLOv11, and the original YOLOv8 baseline, it shows significant mAP improvements of 6.1%, 0.7%, 3.3%, 3.5%, 0.7%, and 0.3%, respectively.

More notably, in a detailed comparison with the YOLOv8 baseline, although the proposed model does not achieve the highest per-class accuracy for white spots, bright lines, and certain surface defects, the performance gap remains minimal. A key advantage of our model lies in its detection of dark line defects, where it achieves a significant performance gain of 1.7%. This improvement is primarily attributed to the GsConv module, which enhances the retention and transmission of weak spatial features characteristic of low-contrast defects like dark lines through its depthwise-separable convolutions and channel shuffling mechanism, effectively preventing these features from being submerged in deeper network layers. This is coupled with the integrated SENetV2 attention mechanism, which adaptively calibrates feature channel weights, significantly amplifying feature responses sensitive to dark line defects while suppressing background noise interference. This targeted enhancement not only directly improves the detection rate for dark lines but also effectively mitigates the performance imbalance across different defect categories observed in the baseline model, thereby collectively contributing to more balanced and robust overall detection performance. Furthermore, the model reduces both the parameter count and computational complexity (GFLOPS), thereby enhancing its practical deployment capability in industrial scenarios without compromising high accuracy.

As shown in Figure 13, the comparative test results of different network models on various defect detection tasks for thermally compressed light guide plates are presented. It can be observed that the YOLOv8n-GSS model demonstrates superior robustness and applicability compared to other networks. Specifically, the detection confidence scores of the proposed model for white spots, bright lines, dark lines, and surface defects reach 0.72, 0.91, 0.85, and 0.73, respectively, representing improvements of 0.01, 0.02, 0.07, and 0.01 over the baseline YOLOv8 network. The experimental results indicate that YOLOv8n-GSS not only performs well on specific datasets but also maintains efficient and reliable detection performance when confronted with diverse industrial defect scenarios. The robustness and generalization capability of the model establish a solid foundation for its practical application in industrial settings, thereby contributing to the intelligent advancement of quality control in manufacturing.

## 5. Limitations and Future Work

The YOLOv8-GSS model still exhibits certain missed detections and false detections on the DO-BSD dataset. To further analyze the phenomena and reasons behind these errors, understand the model’s limitations, and provide direction for subsequent optimization, this paper visualizes some samples with detection failures, and the results are shown in Figure 14.

Figure 14a shows a forging waste defect that was misclassified as a black spot defect, while Figure 14b shows an abrasion defect that was misjudged.

As shown in Figure 14a, the improved network misclassified a large forging waste defect as a black spot defect. This misjudgment primarily stems from two factors: firstly, both types of defects share similar black-color features and variable geometric sizes, resulting in high visual similarity; secondly, the specific forging waste defect far exceeds the conventional size range, almost covering the entire image area, causing it to blend highly with the black background and making distinction difficult. To address this issue, future work could involve targeted expansion of the dataset to include such oversized forging waste defect samples with low contrast against the background, improve the quality of annotations for features like morphology, and guide the model to understand the boundary relationships between defects and the normal background by designing loss functions or network structures.

As shown in Figure 14b, the network exhibited a misjudgment regarding abrasion defects, mistakenly identifying some fine scratches as abrasion defects. This misjudgment arises from two core challenges: firstly, the inherent fine scratches produced during the bearing manufacturing process are highly visually similar to actual abrasion defects, and the complex background texture of the workpiece surface further interferes with judgment; secondly, the vision-based unimodal detection method struggles to perceive and quantify the key physical dimension of scratch “depth,” making it ineffective at distinguishing harmless scratches from functional abrasion. To resolve this issue, subsequent efforts will consider designing channel attention or deformable convolutions to enhance the model’s ability to discriminate subtle feature differences, enabling it to focus more on local micro-textures and depth cues. Alternatively, attempts may be made to integrate 2D visual information with 3D point cloud data or hyperspectral information that can reflect surface contour depth, thereby providing the model with “depth” perception capability and fundamentally overcoming the limitations of unimodal vision.

## 6. Conclusions

A defect detection algorithm based on YOLOv8-GSS was proposed to solve the problems of complex texture background, uneven brightness, defect size, and variety in bearing ring surface images. Specifically, to reduce computational costs, the standard convolution between the C2f module and the Concat module in the neck network is replaced with a lightweight convolution method, namely GsConv. To enhance the model’s diversity and robustness, the SENetV2 attention mechanism is integrated before the SPPF module in the backbone network. The improved YOLOv8-GSS network is used in a large number of experiments on the self-constructed bearing ring defect dataset. The experimental results demonstrate that the proposed YOLOv8-GSS defect detection method achieves an AP_50_ of 97.8%, representing a 1.9% improvement over the YOLOv8 baseline network, while maintaining a detection speed of 115 FPS. It is noteworthy that, compared to the YOLOv8 baseline, our method achieves improvements of 2.4%, 3.6%, and 2.3% in APS, APM, and APL, respectively, while achieving higher inference speed (FPS) with fewer parameters. The improved YOLOv8-GSS network is lightweight while maintaining high real-time detection efficiency to meet the requirements of industrial test applications. Finally, comparative experiments conducted on the hot-pressed light guide plate dataset demonstrate that the proposed improvements exhibit strong generalization capabilities in other industrial defect detection domains. These results not only validate the robustness and adaptability of the method, but also provide valuable insights and potential directions for subsequent research in industrial image detection and processing.

## Figures and Tables

**Figure 1 sensors-25-06504-f001:**
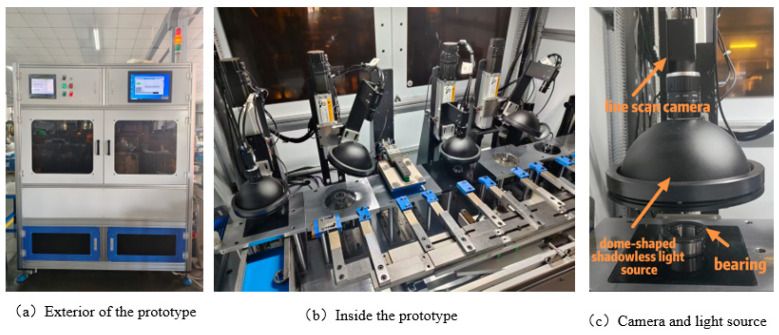
Bearing ring defect detection prototype.

**Figure 2 sensors-25-06504-f002:**
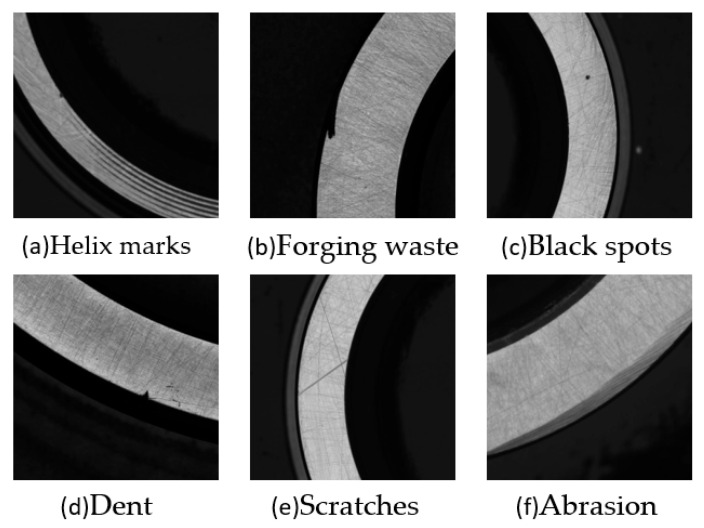
Images of different types of bearing defects.

**Figure 3 sensors-25-06504-f003:**
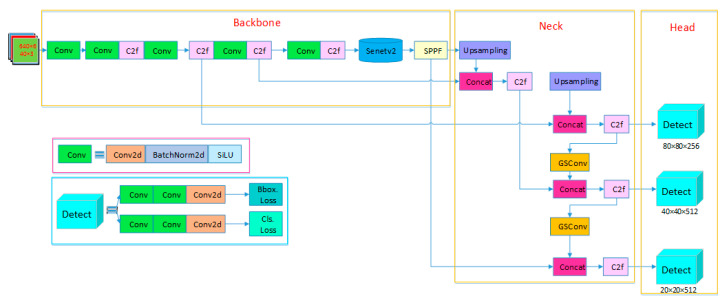
Structure of YOLOv8-GSS.

**Figure 4 sensors-25-06504-f004:**
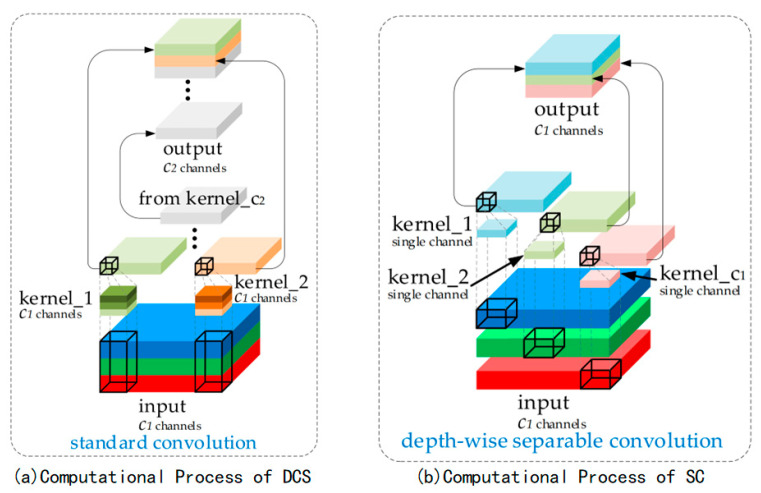
Standard convolution and depth-separable convolution.

**Figure 5 sensors-25-06504-f005:**
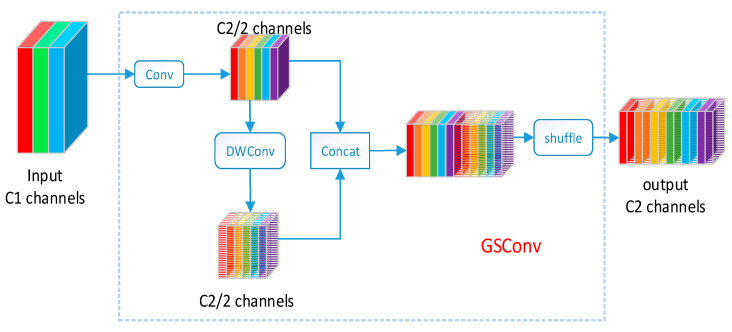
Structure of GSConv.

**Figure 6 sensors-25-06504-f006:**
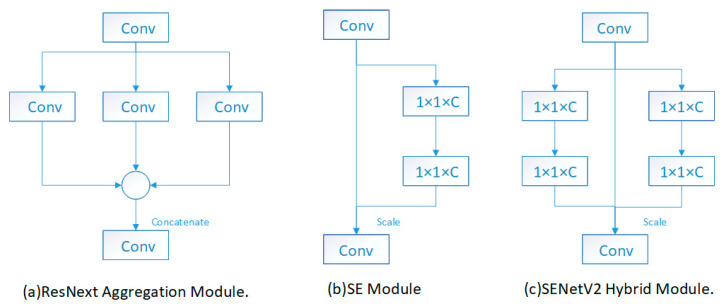
Structure of each module.

**Figure 7 sensors-25-06504-f007:**
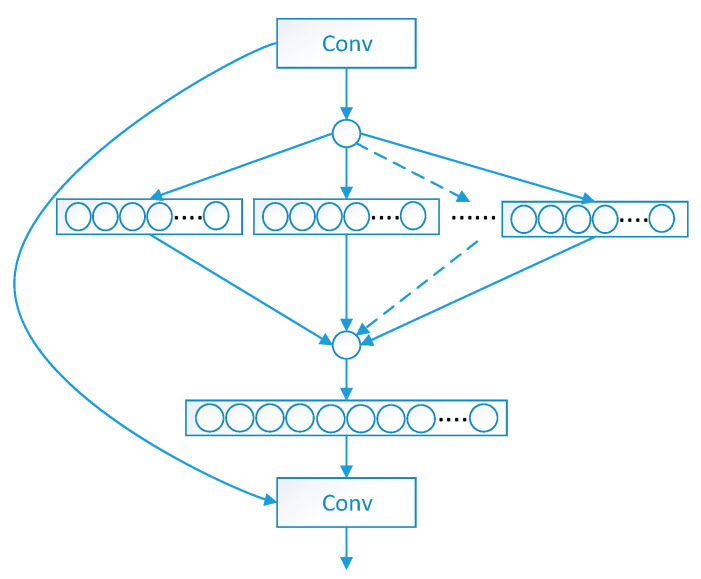
Structure of SaE.

**Figure 8 sensors-25-06504-f008:**
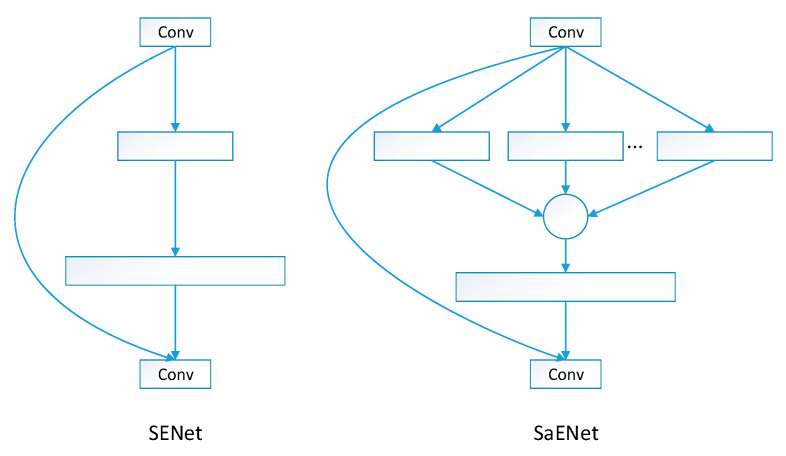
SENet and SENetV2.

**Figure 9 sensors-25-06504-f009:**
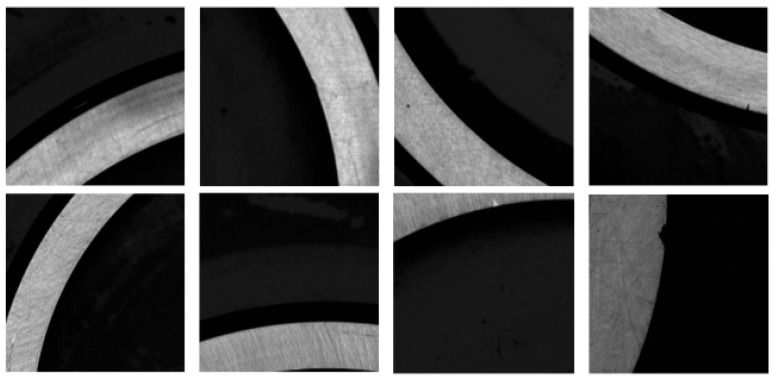
Surface defects of various bearing rings.

**Figure 10 sensors-25-06504-f010:**
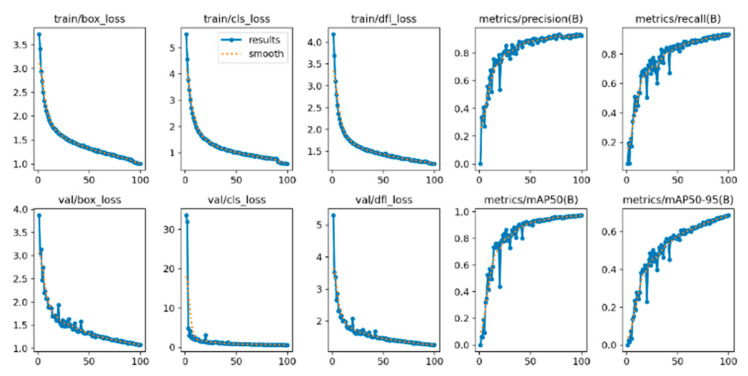
Convergence curves of YOLOv8n on the DO-BSD dataset.

**Figure 11 sensors-25-06504-f011:**
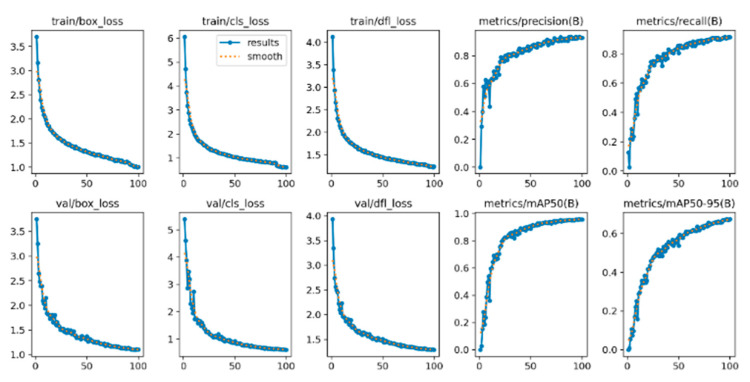
Convergence curves of YOLOv8n-GSS on the DO-BSD dataset.

**Figure 12 sensors-25-06504-f012:**
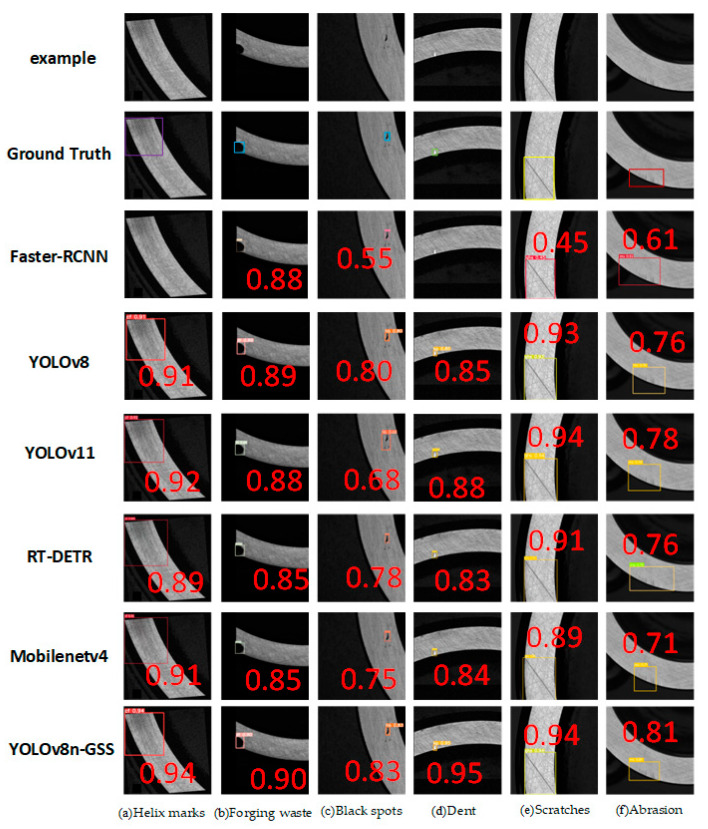
Comparative experimental results on DO-BSD.

**Figure 13 sensors-25-06504-f013:**
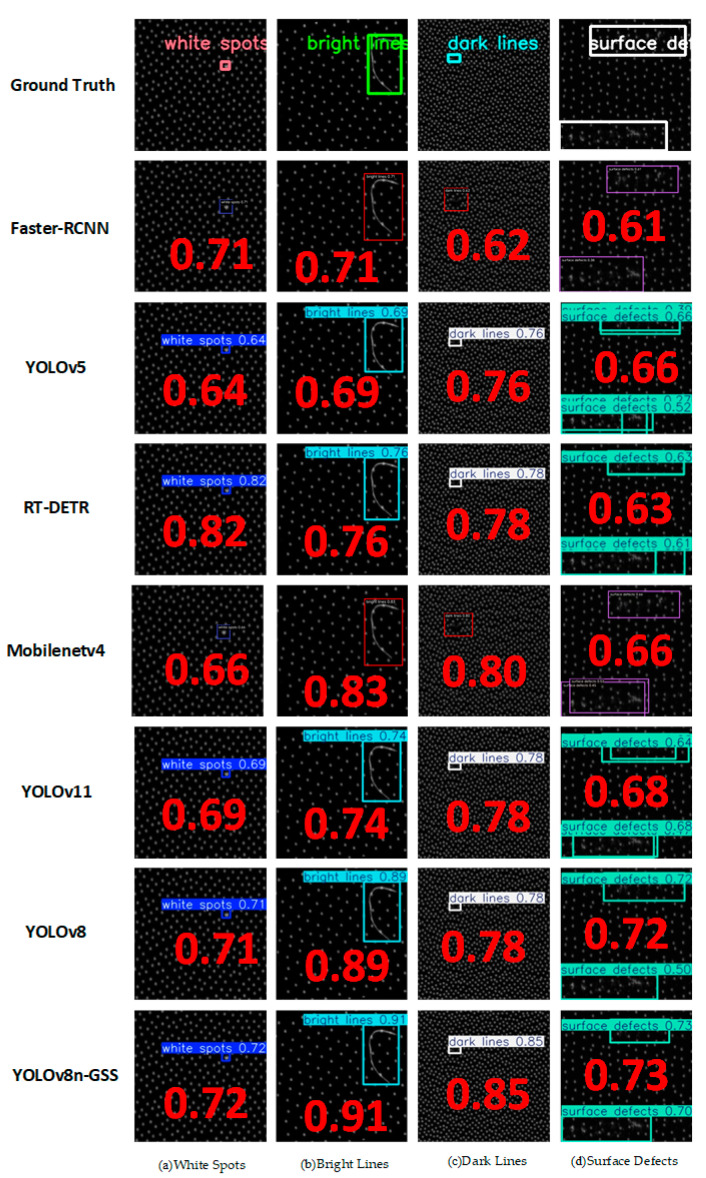
Comparative experimental results on LGP-SDD.

**Figure 14 sensors-25-06504-f014:**
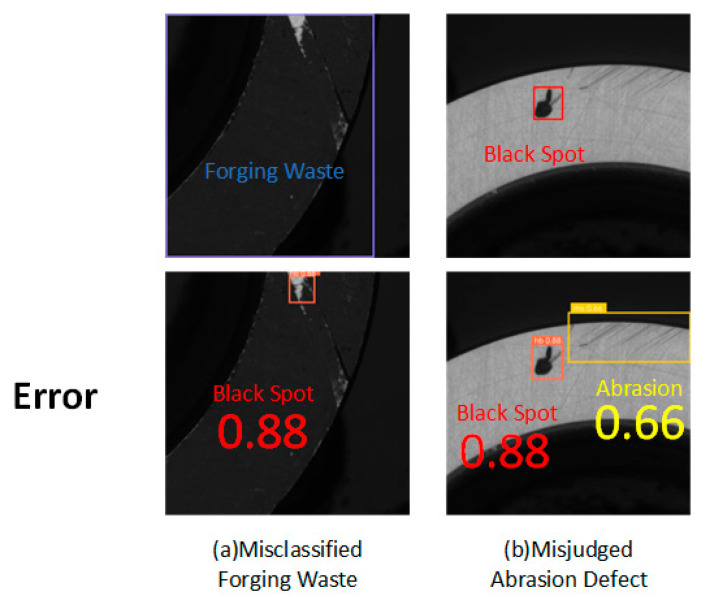
False detection situation.

**Table 1 sensors-25-06504-t001:** Expanded DO-BSD.

	Helix Marks	Forging Waste	Black Spots	Dent	Scratches	Abrasion
Original	570	217	574	280	389	1145
Expansion	1140	1085	1148	1120	1167	1145

**Table 2 sensors-25-06504-t002:** DO-BSD training, validation, and test set statistics.

	Training	Validation	Test	Total
Helix marks	684	228	228	1140
Forging waste	651	217	217	1085
Black spots	689	229	230	1148
Dent	672	224	224	1120
Scratches	700	233	234	1167
Abrasion	687	229	229	1145

**Table 3 sensors-25-06504-t003:** Hardware environment and software version.

	Configuration
Hardware	Operating System: Ubuntu 20.04.06LTS (Linux)
	CPU: Intel(R) Xeon(R) Platinum 8358P (Intel Corporation, Santa Clara, CA, USA)
	RAM: 30G
	GPU: RTX A5000(NVIDIA Corporation; Santa Clara, CA, USA)
Software	Python3.8.13 + Pytorch 1.10 + CUDA11.1

**Table 4 sensors-25-06504-t004:** Network training parameters.

Training Parameter	Value
Batch Size	32
Learning Rate	0.01
Cosine Annealing Learning Rate	0.1
Input Size	640 × 640
Epochs	100

**Table 5 sensors-25-06504-t005:** Results of hyperparameter optimization experiments on YOLOv8-GSS.

Number	Image Size	Epochs	Batch Size	Learning Rate	Optimizer	mAP
1	640	100	32	0.01	SGD	97.8%
2	320	100	32	0.01	SGD	92.1%
3	640	150	32	0.01	SGD	97.6%
4	640	100	32	0.001	RAdam	93.5%
5	640	100	32	0.01	RAdam	94.6%
6	640	100	32	0.1	RAdam	76.1%
7	640	100	16	0.001	SGD	93.0%
8	640	100	16	0.01	SGD	96.5%
9	640	100	16	0.1	SGD	97.5%
10	640	100	32	0.001	SGD	95.8%
11	640	100	32	0.1	SGD	90.9%
12	640	100	64	0.001	SGD	97.5%
13	640	100	64	0.01	SGD	97.0%
14	640	100	64	0.1	SGD	89.5%

**Table 6 sensors-25-06504-t006:** Results of ablation experiments.

Method	GS	SENetV2	mAP	FPS	Parameters	FLOPs(%)
YOLOv8n			95.9%	111	3,011,823	111
GS	√		96.1%	120	2,922,063	112
SENetV2		√	97.2%	107	3,028,207	108
YOLOv8n-Gss	√	√	97.8%	115	2,938,447	115

Note: The check mark (√) indicates that the corresponding module is utilized in the model configuration.

**Table 7 sensors-25-06504-t007:** Performance comparison of object detection methods on DO-BSD.

Models	AP (%)	AP_50_ (%)	AP_75_ (%)	AP_S_ (%)	AP_M_ (%)	AP_L_ (%)	Params (M)	GFLOPs (%)	FPS (Avg)
FasterR-CNN	63.8	83.6	69.2	58.8	47.1	63.4	41.4	208	30
YOLOv5	67.3	97.3	78.9	65.1	61.9	71.7	2.5	7.1	100
RT-DETR	66.9	96.8	77.9	66.3	63.8	74.6	41.94	125.6	65
MobileNetv4	65.2	95.2	74.2	64.2	61.2	70.1	5.6	13.1	125
YOLOv11	68.5	97.2	80.1	66.5	62.1	73	2.6	6.3	115
YOLOv8	68.2	95.9	79.2	65.8	61.2	72.5	3.01	8.1	111
YOLOv8n-GSS	69.4	97.8	82.5	68.2	64.8	74.8	2.93	**7.5**	115

**Table 8 sensors-25-06504-t008:** Comparison of methods on LGP-SDD.

Method	Dataset	AP				mAP (%)	Params (M)	GFLOPs
		White Spots	Bright Lines	Dark Lines	Surface Defects			
FasterR-CNN	LGP-SDD	91.2	95.0	89.3	93.7	92.3	41.4	208
YOLOv5	LGP-SDD	98.5	98.0	95.8	98.7	97.7	2.5	7.1
RT-DETR	LGP-SDD	96.8	95.0	92.5	96.2	95.1	41.9	125.6
MobileNetv4	LGP-SDD	96.1	96	93.2	94.3	94.9	5.6	13.1
YOLOv11	LGP-SDD	98.2	98.2	96.5	97.7	97.7	2.6	6.3
YOLOv8	LGP-SDD	98.8	99.2	95.5	98.8	98.1	3.0	8.1
YOLOv8n-GSS	LGP-SDD	98.7	99.1	97.2	98.6	98.4	2.93	7.5

## Data Availability

The original contributions presented in this study are included in the article. Further inquiries can be directed to the corresponding author.

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
