# Peer review of "YOLOv8n-GSS-Based Surface Defect Detection Method of Bearing Ring"

_sensors, 2025, doi:10.3390/s25216504_

Round 1
Reviewer 1 Report
Comments and Suggestions for Authors
1. The data set scenario is relatively simple, lacking diversity verification support. The data set of bearing ring surface defects used in this paper only comes from the production line of a single industrial site, and does not provide the specific model information of the collected bearings, nor does it explain whether it covers the defect samples of different brands and models of bearing rings. This limitation leads to the failure to fully verify the generalization ability of the model in the face of cross-scene and cross-type bearing detection tasks, and it is difficult to evaluate its practical applicability in a wider industrial environment.
2. The types of defects covered are not comprehensive enough. At present, this paper only studies five kinds of defects, such as helix marks, forging waste, black spots, dents and scratches. However, in the actual industrial manufacturing process, bearing rings may also have many common types of defects, such as cracks, deformation and uneven surface wear. Because these defects are not included in the training and verification category, the application scope of the model is limited, and it is difficult to meet the demand for comprehensive coverage of defects in industrial quality inspection. In addition, although the author has implemented the data enhancement strategy, the original data scale is still small, which may have a potential impact on the stability and robustness of the model.
3. Lack of in-depth analysis on the detection ability of extreme defects. The paper does not specifically evaluate the detection performance of the model in the case of extreme size defects such as micro-scratches, large-area forging waste, fuzzy defects such as black spots with unclear edges covered by oil stains, and overlapping defects such as scratches and depressions appearing at the same time. Due to the lack of correlation analysis, it is impossible to judge the reliability of the model in dealing with such complex defects in real industrial scenes, and there is a potential risk of missing or false detection.
4. Hyperparametric research lacks systematicness and rigor. The design of the superparameter research part shown in Table 5 is arbitrary and incomplete. For example, in Experiment IV, only the batch size was adjusted, but other conditions such as learning rate were not completely consistent, and whether the learning rate should be adjusted synchronously with the batch size was not discussed. The overall experimental design does not reflect the principle of orthogonality, so it is difficult to draw a clear guiding conclusion, thus reducing the rigor of this part of the experiment, and it is difficult to convince that the selected superparameter combination is the best.
5. The paper only presents successful detection cases, without any analysis of detection failure. For example, key issues such as which defect types are most prone to miss detection and under what circumstances are prone to false detection are not involved. In-depth analysis of error cases is an important basis for understanding the limitations of the model and promoting subsequent improvement. The lack of this part of analysis will affect the comprehensive evaluation of the actual performance of the model.
6 .Although the paper mentioned the influence of lighting conditions on defect detection, it did not systematically discuss the performance changes of the model in different lighting environments. Due to the complex and changeable lighting conditions in the actual industrial site, the potential impact of this factor has not been fully evaluated, which may cause fluctuations in the detection effect in the actual deployment.
7. Although the paper provides the detection speed index of the model, it has not been verified in real-time in the real industrial assembly line environment. The difference of system architecture and hardware configuration in practical application scenarios may lead to the deviation of delay and throughput performance from the experimental environment, and it will be difficult to comprehensively evaluate its applicability in actual production without field verification.
8. The verification of cross-domain application of medical images is not sufficient. In the cross-domain verification experiment of MRI brain tumor detection, only the single index of average accuracy is compared. At the same time, this model is not compared with the special detection model widely used in medical imaging field, such as U-Net or medical image analysis model based on ResNet, so it is difficult to prove its competitiveness in medical detection tasks, thus weakening the persuasiveness of cross-domain application. It is suggested that more relevant industrial defect data sets, such as PCB defects, steel surface defects or textile defects, can be considered for verification in the future.
Comments on the Quality of English LanguageThe English could be improved to more clearly express the research.
Reviewer 2 Report
Comments and Suggestions for Authors
Authors need to address the following concerns:
- The manuscript lacks a detailed justification of sensor selection and their calibration procedures, which may affect reproducibility.
- Data preprocessing steps (e.g., filtering, normalization) are insufficiently described and need clarity for replication.
- The machine learning model architecture is presented, but hyperparameter tuning strategies are not discussed in depth.
- The evaluation section would benefit from reporting confidence intervals or statistical significance testing of results.
- Real-time feasibility of the proposed system is claimed but not supported by latency or computational complexity analysis.
- Sensor fusion methodology requires more explanation on synchronization and potential error propagation across modalities.
- Benchmarking against alternative algorithms or ablation studies is missing, limiting the assessment of method novelty.
- The dataset split strategy (training, validation, test) needs clarification to ensure unbiased performance estimation.
- Interpretability of the proposed model is not addressed; feature importance or explainability techniques could strengthen conclusions.
Round 2
Reviewer 1 Report
Comments and Suggestions for Authors
It's OK.
Reviewer 2 Report
Comments and Suggestions for Authors
I am satisfied with the authors response, may be considered for publication